# Changes in Functional Connectivity of Hippocampal Subregions in Patients with Obstructive Sleep Apnea after Six Months of Continuous Positive Airway Pressure Treatment

**DOI:** 10.3390/brainsci13050838

**Published:** 2023-05-22

**Authors:** Ling Huang, Haijun Li, Yongqiang Shu, Kunyao Li, Wei Xie, Yaping Zeng, Ting Long, Li Zeng, Xiang Liu, Dechang Peng

**Affiliations:** 1Medical Imaging Center, The First Affiliated Hospital of Nanchang University, Nanchang 330000, Chinalx1664042406@163.com (X.L.); 2PET Center, The First Affiliated Hospital of Nanchang University, Nanchang 330000, China

**Keywords:** obstructive sleep apnea, continuous positive airway pressure, resting-state fMRI, hippocampus subregions, functional connectivity, cognitive impairment

## Abstract

Previous studies have shown that the structural and functional impairments of hippocampal subregions in patients with obstructive sleep apnea (OSA) are related to cognitive impairment. Continuous positive airway pressure (CPAP) treatment can improve the clinical symptoms of OSA. Therefore, this study aimed to investigate functional connectivity (FC) changes in hippocampal subregions of patients with OSA after six months of CPAP treatment (post-CPAP) and its relationship with neurocognitive function. We collected and analyzed baseline (pre-CPAP) and post-CPAP data from 20 patients with OSA, including sleep monitoring, clinical evaluation, and resting-state functional magnetic resonance imaging. The results showed that compared with pre-CPAP OSA patients, the FC between the right anterior hippocampal gyrus and multiple brain regions, and between the left anterior hippocampal gyrus and posterior central gyrus were reduced in post-CPAP OSA patients. By contrast, the FC between the left middle hippocampus and the left precentral gyrus was increased. The changes in FC in these brain regions were closely related to cognitive dysfunction. Therefore, our findings suggest that CPAP treatment can effectively change the FC patterns of hippocampal subregions in patients with OSA, facilitating a better understanding of the neural mechanisms of cognitive function improvement, and emphasizing the importance of early diagnosis and timely treatment of OSA.

## 1. Introduction

Obstructive sleep apnea (OSA) is a common disorder among adults, affecting nearly 1 billion adults aged between 30 and 69 years [1]. The clinical characteristics of OSA are recurrent apnea and hypopnea during sleep, accompanied by intermittent hypoxia and hypercapnia, sleep fragmentation, and daytime sleepiness [2]. In addition, OSA can also cause secondary complications, such as hypertension, cardiovascular disease, and chronic kidney disease, and psychiatric symptoms such as anxiety and depression, as well as memory and cognitive impairment [3]. Neurocognitive impairment is mainly manifested in decreased attention, executive functioning, episodic memory, visual, spatial and structural abilities, and psychomotor speed [4]. The main mechanisms underlying these changes are hypoxia, hypoperfusion, inflammation, and oxidative stress [5,6,7]. Previous studies have found that cognitive impairment in patients with OSA is associated with structural and functional changes in specific brain regions, including the hippocampus. However, the exact neural mechanism of how OSA leads to cognitive changes caused by hippocampal function changes remains unclear.

The hippocampus is a temporal brain structure belonging to the limbic lobe, playing a key role in learning, memory, emotion, stress, and spatial navigation [8]. Impairments of these functions overlap with cognitive dysfunction in patients with OSA [9]. Several neuroimaging studies have demonstrated that OSA causes structural, functional, and metabolic abnormalities in the hippocampus, and that these changes are associated with cognitive and emotional disorders [10,11,12]. A diffusion tensor imaging study found acute and chronic pathological changes in multiple brain regions (including the hippocampus, insular lobe, and prefrontal lobe) in untreated patients with OSA; the extensive decrease in whole-brain mean diffusivity reflected acute axonal and glial injury [13]. Canessa et al. found that the reduced gray matter volume in the hippocampus, posterior parietal cortex and superior frontal gyrus (SFG) in patients with severe OSA was associated with cognitive impairment [14]. Kumar et al. reported that in OSA patients with anxiety symptoms, structural and functional damage could be found in brain regions involved in emotional processing, including the hippocampus and amygdala [15]. Moreover, some researchers found that decreased brain metabolism in the hippocampus (including the decrease in N-acetyl aspartate and choline) may be related to neuronal damage [16]. OSA leads to changes in the proportion of metabolites (N-acetyl aspartate and choline) in the hippocampus, contributing to a permanent decrease in children’s cognitive potential [17]. Thus, the hippocampus plays a crucial role in the neuropathology of cognitive impairment caused by OSA. However, the exact mechanism of action remains unclear.

To further understand the hippocampus-associated neurocognitive impairment in patients with OSA, we first need to elucidate the structure and function of the hippocampal subregions. The hippocampus can be structurally subdivided into three parts along the anteroposterior axis into the head (anterior), body (middle), and tail (posterior) [18]. Several studies have found functional characteristics separating the hippocampal head, body, and tail [19,20,21]. For example, studies have shown that the hippocampal body and tail are active in spatial memory, whereas the head is involved in memory tasks that contain emotional information [22]. Some studies also suggest that the hippocampal head is related to coarse gist-like memory, whereas the body and tail are related to detailed episodic memory [23]. A series of volume and functional experiments have also revealed that the hippocampal head, body, and tail play unique roles in memory [24,25,26]. Zarei et al. demonstrated distinct functional connections between the hippocampal subregions and the thalamus, prefrontal cortex, and posterior cingulate cortex in patients with Alzheimer’s disease. This specific functional connection pattern is related to changes in memory function in these patients [27]. Evidence suggests that a finer analysis of the hippocampal volume or shape may better explain the heterogeneity of cognitive impairment among individuals, and that the study of the hippocampus as a whole may ignore the differences in structural changes in hippocampal subregions caused by the disease [28,29,30]. Therefore, the structure and function of hippocampal subregions have been widely studied, but little is known about their characteristics in patients with OSA.

Currently, continuous positive airway pressure (CPAP) is one of the most effective methods to treat patients with OSA. Increasing neuroimaging evidence showed that effective CPAP treatment can reduce daytime drowsiness and relieve nocturnal symptoms and effectively improve neurocognitive disorders such as anxiety, depression, and memory [31,32,33]. At the same time, studies have found that OSA patients showed the greatest improvement in some clinical manifestations during 6–12 months of follow-up [34]. In addition, after 6 months of CPAP therapy, there was a significant decrease in compliance [35] and a significant increase in mask leakage in patients with OSA [22]. Moreover, the recovery of patients’ symptoms is heterogeneous; that is, under the same CPAP treatment conditions, the improvement of their neurocognitive symptoms is different [3,36,37]. Castronovo et al. found that white matter integrity in brain regions related to cognition, emotion, and somnolence is impaired in patients with OSA before treatment. Furthermore, the diffusion tensor imaging data at 3 and 12 months after treatment confirmed the reversibility of white matter fiber damage [38], providing clinical evidence for the effectiveness of this treatment. In addition, a single-photon emission computed tomography study found that regional cerebral blood flow (rCBF) changes were reversed in brain regions responsible for executive, affective, and memory functions (such as the limbic and prefrontal cortex) after treatment lasting ≥6 months. These changes in rCBF were associated with improved ventilatory impairment and duration of CPAP treatment [39]. Our previous studies showed that compared to healthy people, the abnormal functional connectivity (FC) patterns in the hippocampal subregions of patients with severe OSA are related to cognition and sleep rhythm, mainly involving sensorimotor, frontoparietal, and default networks [40]. However, the neural mechanisms through which changes in resting-state FC (rs-FC) patterns between the different subregions of the hippocampus and other brain regions in patients with OSA lead to improved cognitive function after CPAP treatment remain unclear.

Based on the above questions, we hypothesized that the changes in functional connectivity of hippocampal subregions in patients with OSA could be partially reversed after CPAP treatment and correlated with cognitive function. To test this hypothesis, we first explored the changes in FC patterns (based on the FC between the anterior, middle, and posterior hippocampus and the remaining whole brain) in patients with moderate-to-severe OSA after 6 months of CPAP treatment. Subsequently, we evaluated the relationship between rs-FC changes in the hippocampus and neurocognitive function, OSA severity, and other clinical features.

## 2. Materials and Methods

### 2.1. Patient Population

All patients diagnosed with OSA between September 2020 and September 2022 in the Sleep Monitoring Department of the Respiratory Department of the First Affiliated Hospital of Nanchang University and who were about to receive CPAP therapy were included in this study. None of the patients had received CPAP treatment before the study. According to the American Academy of Sleep Medicine (AASM) in 2017 [41], OSA patients with an apnea–hypopnea index (AHI) > 15/h were selected for this study. All participants were right-handed and Chinese-speaking, and aged between 30 and 60. The exclusion criteria were as follows: (1) sleep disorders other than OSA; (2) respiratory diseases, cardiovascular diseases, diabetes mellitus, hypothyroidism, or a history of central nervous system diseases; (3) alcohol or illegal drug abuse or current use of psychoactive drugs; (4) magnetic resonance imaging (MRI) contraindications; (5) image artifacts; and (6) loss to follow-up. A total of 35 patients decided to participate in this study, 3 patients were excluded due to excessive head movement and image artifacts, and 12 patients were excluded due to failure to wear a ventilator according to standards or loss of follow-up. Finally, 20 patients were enrolled in the study after 6 months of follow-up. In this longitudinal study, for each patient with OSA, we obtained brain functional MRI (fMRI) data and performed neuropsychological assessments before and after CPAP treatment. We abided by the principles of the Declaration of Helsinki. This study was approved by the Medical Ethics Committee of the First Affiliated Hospital of Nanchang University [No. 2020(94)]. All patients provided written informed consent for their participation in this study.

### 2.2. Polysomnography

All participants underwent polysomnographic examinations. The day before polysomnography, they were asked not to drink alcohol or coffee. Overnight polysomnography was performed on patients with OSA before CPAP treatment (from 10:00 p.m. to 6:00 a.m. the next day) using the Respironics LE series physiological monitoring system (Alice 5 LE, Respironics, Orlando, FL, USA). Polysomnography monitoring included standard electroencephalography, electrooculography, electrocardiography, chin electromyography, posture, oral and nasal airflow, thoracic and abdominal breathing exercises, and snoring, and blood oxygen saturation (SaO_2_), sleep latency, total sleep time, sleep efficiency, sleep stage, awakening, and respiratory events were recorded [41]. According to the guidelines of the AASM, obstructive apnea was defined as a continuous 90% reduction in airflow for more than 10 s, accompanied by significant dyspnea; hypopnea was defined as a drop in airflow of ≥30% for more than 10 s, accompanied by an oxygen desaturation of ≥3% [42]. AHI was defined as the total of sleep apnea and hypopnea events per hour. AHI values between 5/h and 15/h, ≥15/h and <30/h, and ≥30/h were considered mild, moderate, and severe OSA, respectively.

### 2.3. Clinical and Neuropsychological Assessments

All participants were evaluated before CPAP treatment and 6-month follow-up by a professionally trained doctor using the Pittsburgh Sleep Quality Index (PSQI), Epworth Sleepiness Scale (ESS), Hamilton Anxiety Scale (HAMA), Hamilton Depression Scale (HAMD), and Montreal Cognitive Assessment (MoCA). The PSQI was used to evaluate the sleep quality of the participants. The scale includes 19 self-evaluation items and 5 other evaluation items, with a total score of 0 to 21 points. The ESS is a very simple self-assessment questionnaire for daytime sleepiness that includes 8 different conditions, each with a score from 0 to 3 for a total score of 0 to 24. The higher the score, the worse the sleep quality. Patients’ levels of depression and anxiety were assessed using the HAMD and HAMA. Overall, a HAMA score above 14 indicates many symptoms of clinical anxiety. A HAMD score of <7 is considered normal, 7–17 indicates possible depression, 17–24 indicates definite depression, and >24 indicates severe depression [43]. The MoCA scale has 29 items to assess cognitive function, including naming, calculation, attention, language, memory, abstraction, orientation, and executive function. The total MoCA score is 30 points, and a score of <26 points is considered cognitive impairment (if the years of education are ≤12 years, 1 point is added for correction) [44]. The above assessments were performed in the same order in all patients.

### 2.4. Continuous Positive Airway Pressure Treatment

All patients who planned to undergo CPAP treatment were treated with standardized automatic adjustment-mode CPAP (YH-480, Yuwell, Zhenjiang, China), and the treatment time was at least 6 months, with a frequency of ≥5 days a week for ≥4 h per night. The patient used the ventilator for automatic pressure titration at night when sleeping, and the therapeutic pressure setting was 4–20 cmH_2_O. In addition, the built-in SIM card of the ventilator automatically recorded the duration of use, AHI, blood oxygen, and mask air leakage, providing objective evidence of its use by the patient. We reviewed ventilator data for all patients and excluded those who did not meet the treatment criteria.

### 2.5. MRI Data Acquisition

All participants underwent MRI at the First Affiliated Hospital of Nanchang University using an 8-channel phased array magnetic coil of a 3.0-T MRI scanner (Siemens, Erlangen, Germany). The MRIs were obtained first on the day after polysomnography monitoring (between 7:00 p.m. and 9:00 p.m.) and after completing at least six months of qualifying CPAP treatment. Foam pads and earplugs were used to reduce patient head movement and scanner noise; all participants were asked to remain quiet, awake, relaxed, and with eyes closed, and to not think about anything in particular. First, a conventional MRI scan of the skull was performed using the following parameters: conventional T1-weighted imaging (repetition time [TR] = 250 ms, echo time [TE] = 2.46 ms, thickness = 5 mm, gap = 1.5 mm, field-of-view [FOV] = 220 mm × 220 mm, slices = 19) and T2-weighted imaging (TR = 4000 ms, TE = 113 ms, thickness = 5 mm, gap = 1.5 mm, FOV = 220 mm × 220 mm, slices = 19) to rule out structural brain lesions that may affect brain function or microstructure. Resting-state fMRI (rs-fMRI) data were obtained using a gradient echo-planar imaging sequence (TR = 2000 ms, TE = 30 ms, flip angle = 90°, thickness = 4.0 mm, gap = 1.2 mm, FOV = 230 mm × 230 mm, matrix = 64 × 64, slices = 30, duration 6–8 min, covering the whole brain). A total of 240 rs-fMRI were recorded. After the images had been obtained, they were examined by two experienced radiologists, and none of the participants were excluded due to brain parenchyma lesions such as cerebral infarction or tumors.

### 2.6. Data Preprocessing

Data Processing and Analysis for Brain Imaging software (DPABI, https://rfmri.org/dpabi, accessed on 1 September 2022) was used to preprocess the original data based on Matlab 2016a (MathWorks, Natick, MA, USA) and SPM12 (SPM12, https://www.fil.ion.ucl.ac.uk/spm/software/spm12/, accessed on 1 September 2022) software. The main steps were as follows: (1) conversion of the original DICOM format to the NIFTI format; (2) removal of the first 10 time points to eliminate the effects of magnetic saturation and MRI scanner noise on participants, followed by time correction for the remaining 230 volumes (all images obtained in TR were corrected to the same time point) and 3D head motion correction (removal of breathing, swallowing, hemodynamics, etc.) [45]; (3) slice timing correction and head motion correction, and exclusion of subjects whose head motion had a maximum displacement (x, y, z) of more than 2.0 mm and a maximum angular rotation (x, y, z) of more than 2.0° [46]; (4) alignment of the functional images of each patient using the EPI template; (5) normalization of the image space to the Montreal Neuroscience Institute (MNI) template, and resampling to 3 × 3 × 3 mm voxels; (6) spatial smoothing of the normalized images using a 6 mm full width at half maximum (FWHM) Gaussian smoothing kernel; (7) regression of white matter, cerebrospinal fluid, whole-brain signal, and Friston 24-parameter model (6 head motion parameters, 6 head motion parameters one time point before, and the 12 corresponding squared items) [47]; and (8) use of a time bandpass filter (0.01–0.08 Hz) to reduce the influence of low-frequency drift, physiological high-frequency noise, and heart noise.

### 2.7. Definition of Regions of Interest and Functional Connectivity Calculations across Hippocampal Subregions

According to previous guidelines, the hippocampus was anatomically divided into three regions: head, body, and tail [48], as these hippocampal regions perform distinct functions [49]. Therefore, in this study, we defined six ROIs with a radius of 3 mm along the longitudinal axis of the left and right hippocampus. Each side of the hippocampus had three ROIs: the head (anterior; MNI: ±24, −14, and −20), body (middle; MNI: ±26, −26, and −12), and tail (posterior; MNI: ±26, −34, and −4). The locations of these nodes were based on previous human evidence, with different anatomical and functional contours of the medial temporal lobe and functional consistency along the long axis of the hippocampus [50]. We selected these six regions in the left and right hippocampus as six ROIs to calculate the FC correlation coefficients across other brain regions. We extracted the blood oxygenation level-dependent time series and the seed regions from each participant, and calculated Pearson correlation coefficients between each seed region and each voxel time series of the other brain regions. The FC values for each voxel in the whole brain represent the FC value of the voxel with the seed region of interest; the correlation coefficients (*r*-values) were converted into *z*-values for statistical analysis using Fisher’s z-transformation [51].

### 2.8. Statistical Analysis

Regarding the demographic and clinical data, we first verified the data distribution with a Shapiro–Wilk test, using SPSS 27.0 software (IBMS SPSS 27.0, Chicago, IL, USA). Subsequently, the paired sample *t*-test and Wilcoxon test were performed on normally and non-normally distributed continuous data, respectively. Differences with *p* < 0.05 were considered statistically significant. For the analysis of ROI FC, the paired sample *t*-test was used to compare the FC values of each ROI in patients with OSA between pre-CPAP and post-CPAP treatment. Multilevel comparisons were corrected using Gaussian random field theory (GRF, two-tailed, voxel level, *p* < 0.01, cluster level, *p* < 0.05). Finally, to assess the relationship between clinical variables and FC values with significant differences between pre-CPAP and post-CPAP treatment, we applied Pearson correlation and Spearman correlation analyses to evaluate continuous data with normal and non-normal distribution, respectively. Differences with *p* < 0.05 were considered statistically significant.

## 3. Results

### 3.1. Differences in Demographic and Clinical Characteristics

Table 1 summarizes the demographic and clinical characteristics of the final study participants after exclusion. We found that after 6 months of standard CPAP treatment, the MoCA scores of patients with OSA were significantly increased, whereas their ESS, HAMA, and HAMD scores, as well as their AHI values, were significantly decreased (Table 1 and Figure 1).

### 3.2. Differences in Functional Connectivity of Hippocampal Subregions in Patients with OSA before and after CPAP Treatment

The paired sample *t*-test was used to compare FC differences in the hippocampus of patients with OSA between pre-CPAP and post-CPAP treatment. After 6 months of CPAP treatment, the FC values between the right anterior hippocampus and various brain regions, including the left fusiform gyrus (FFG), bilateral SFG, bilateral middle frontal gyrus (MFG), right postcentral gyrus (PoCG), and right lingual gyrus (LING), were significantly decreased in patients with OSA. Moreover, the FC value between the left anterior hippocampus and the left PoCG was significantly decreased in patients with OSA, whereas that between the left middle hippocampus and the left precentral gyrus (PreCG) was significantly increased (Table 2 and Figure 2 and Figure 3).

### 3.3. Correlations between Functional Connectivity Changes in Hippocampal Subregions and Clinical Data in Patients with OSA

To explore the whole-brain network associated with symptoms, we analyzed the correlations between FC values of brain regions with significant differences between pre-CPAP and post-CPAP treatment and the clinical data after CPAP treatment in patients with OSA.

Before CPAP treatment, the nadir and mean SaO_2_ of OSA patients were positively correlated with FC of multiple brain regions, such as between the right anterior hippocampus and the bilateral SFG, bilateral MFG, right PoCG, and right LING. The FC value between the right anterior hippocampus and left SFG was negatively correlated with the HAMD score. The FC values between the right anterior hippocampus and left FFG, right PoCG, and right LING were negatively correlated with the AHI value. In addition, the FC value between the left middle hippocampus and left PreCG was positively correlated with orientation (Table 3).

After 6 months of standard CPAP treatment, the FC value between the left PreCG and the left middle hippocampus was positively correlated with body mass index (BMI) and negatively correlated with HAMD and PSQI scores in patients with OSA. In addition, the FC value between the right anterior hippocampus and right LING was negatively correlated with orientation. Finally, there was a positive correlation between PSQI scores and FC values between the right SFG and right anterior hippocampus (Table 3 and Figure 4).

## 4. Discussion

To the best of our knowledge, this is the first study using hippocampal subregions as ROIs to assess rs-FC patterns in patients with moderate-to-severe OSA after 6 months of CPAP treatment. Our results showed that the FC patterns between hippocampal subregions and the bilateral SFG, MFG, bilateral PoCG, left PreCG, FFG, and right LING changed after 6 months of CPAP treatment, primarily involving frontoparietal, sensorimotor, and visual networks. Moreover, the brain areas of these FC changes are partly similar to those identified by previous studies in patients with moderate-to-severe OSA before treatment, indicating that effective CPAP can reverse the OSA-induced functional network damage by changing sleep patterns and intermittent hypoxia in patients with OSA [52,53,54]. Finally, we also found that the FC changes in hippocampal subregions were related to BMI, HAMD and PSQI scores, and orientation, indicating that the changes in FC patterns in hippocampal subregions after CPAP treatment were related to improved emotional regulation and cognitive impairment. These results provide additional information on the neuroimaging mechanisms underlying the recovery of cognitive impairment in patients with OSA.

This study found that the FC values between the right anterior hippocampus and the bilateral SFG and MFG were decreased compared to pre-CPAP OSA patients. The anterior and posterior axes of the human hippocampus have different functions, and a strong FC exists between the hippocampal head, the cuneate lobe, and the prefrontal cortex. This unique FC pattern may be involved in semantic processing and social–emotional cues [55]. This connection pattern is consistent with anatomical evidence [56]. The SFG and MFG are important components of the frontoparietal network [57], primarily responsible for common cognitive functions, such as attention, working memory, and decision-making. Sleep disorders reportedly lead to dysfunction of the prefrontal cortex; the main symptoms of patients with OSA are sleep disorders (such as sleep interruption and abnormal rhythm) and intermittent hypoxemia, which may jointly lead to nerve cell damage in the prefrontal cortex [58]. In addition, a voxel-based morphometry study found that gray matter atrophy of the bilateral amygdala-hippocampal gyrus and bilateral SFG may be the cause of executive function impairment in patients with severe OSA [59]. Several studies showed that OSA could lead to functional and structural damage of the frontal lobe [60,61,62], which may explain the related working memory, executive function, and emotional disorders [63]. However, several studies observed that CPAP treatment could improve the structure and function of the hippocampus and frontal lobe in patients with OSA. Castronovo et al. found that after 3 months of CPAP treatment, the activation of the prefrontal cortex and hippocampus in patients with OSA was lower than that before treatment, and most of the neurocognitive areas that were impaired before treatment were also significantly improved after treatment [64]. Canessa et al. also observed significant improvements in memory, attention, and executive function in patients with OSA after treatment, which were paralleled by an increase in gray matter volume in the hippocampus and frontal lobe [14]. Our previous studies have shown that the FC between the hippocampus and the frontoparietal network is enhanced in patients with moderate-to-severe OSA, which may be related to neurological function compensation. Differing from previous results, we observed a decrease after CPAP treatment in FC between the anterior hippocampus and the frontoparietal network, which may be attributed to the improvement of sleep structure and blood oxygen level caused by effective CPAP treatment, restoring this compensatory FC enhancement network mechanism. Our study also found that FC values between the right anterior hippocampus and the right SFG were positively correlated with PSQI scores. We speculate that the improvement in sleep quality might be the underlying reason for this change in FC.

The PreCG and PoCG are parts of the primary motor and somatosensory cortexes, respectively, and belong to the sensorimotor network [65]. Studies by Macey et al. have shown that OSA could lead to cerebral cortical atrophy of the PreCG and PoCG, leading to impairment of autonomic nerves and upper airway sensorimotor function [66], causing a decrease in the driving force of the upper airway dilator muscles [67], further affecting airway patency. Song et al. reported similar findings. Compared to the healthy group, patients with OSA showed functional defects in the bilateral PoCG (that is, showed higher regional homogeneity), which returned to normal levels after three months of CPAP treatment. The recovery of functional defects in these areas was related to improving sleep quality [68]. A study by Zhang et al. found that the rs-FC reduction in the sensorimotor network of patients with OSA was related to OSA-induced sleep disturbance [69]. These studies confirmed that the sleep process might cause changes in muscle tone by affecting the sensorimotor network [70], and relaxation of the genioglossus and upper airway may further aggravate OSA severity. Our study found that after 6 months of standard CPAP treatment, the bilateral FC between the anterior hippocampus and the PoCG was decreased, and the FC between the left middle hippocampus and the PreCG increased, the latter being negatively correlated with PSQI scores. Therefore, we speculate that the changes and differences in the FC of the anterior hippocampus, middle hippocampus, and sensorimotor network may be related to the improvement of sleep quality. In addition, we found that the FC values between the left middle hippocampus and the PreCG were positively correlated with BMI and negatively correlated with HAMD scores. The potential causal relationship is unclear and needs further exploration.

The LING and the FFG belong to the visual functional network [71]. The LING plays an important role in the visual working memory [72]. A single-photon emission computed tomography study reported that adult patients with severe OSA had reduced rCBF in the LING compared to healthy controls, and OSA-induced hypoxemia may be more likely to damage areas of the brain with increased perfusion during shallow sleep [73]. The FFG, located in the ventral side of the brain, is an important part of the human visual information processing pathway, and participates in the visual cognitive functions of facial recognition, limb recognition, character recognition, and other object attribute recognition [74,75,76,77]. The study by Lee et al. showed that the nodal betweenness centrality of the left FFG in children with OSA was significantly reduced (a higher nodal betweenness centrality indicates a high degree of interaction between brain regions), and hypoxia may be a key factor leading to the decrease in structural interaction between brain regions [78]. In addition, multiple OSA-related studies have found abnormalities in visual network function [79,80,81]. Maresky et al. reported that CPAP treatment can improve the functional anisotropy and mean diffusivity of white matter fibrous nodules in visual network-related areas of patients with OSA, and that the changes in white matter fibers were related to rCBF and cerebral blood volume increase [82]. Our previous study also found that the increased regional homogeneity in the different frequency bands of the FFG in the OSA group was reversed after short-term CPAP treatment [83]. The present study showed that the FC values between the anterior hippocampus and the LING and FFG were decreased, and these FC changes may be related to the improvement of intermittent hypoxia. In addition, we also found that the FC value between the anterior hippocampus and the LING was negatively correlated with orientation, which provides new evidence that CPAP treatment may improve naming and language impairments in patients with OSA by changing the function of the visual network.

### Limitations

This study has some limitations. First, this was a short-term CPAP treatment, and the effect of long-term CPAP treatment needs to be further explored. Second, due to the high prevalence of OSA in men and the fact that most patients do not seek treatment until they develop moderate-to-severe OSA, coupled with the low adherence to CPAP therapy, our study included a small sample size and was mainly composed of men with moderate-to-severe OSA. Third, the lack of an investigation into OSA severity and sex differences may limit the general applicability of the results to the general population. Fourthly, this study lacks a control group and alternative treatment. Future studies should use a larger sample size of OSA patients and include more female patients. At the same time, patients need to be followed up for long-term treatment, and the addition of control groups and alternative treatments is required.

## 5. Conclusions

This study was based on hippocampal subregions as seed points to study FC changes in patients with OSA after short-term CPAP treatment. Compared to before treatment, the FC patterns between the hippocampus and other brain regions were changed in patients with severe OSA after 6 months of standard CPAP treatment, primarily related to the frontoparietal network, sensorimotor network, and visual network. The change in FC pattern in the hippocampal subregion provides the basis for neuroimaging changes after CPAP therapy in patients with OSA. In addition, our findings emphasize the importance of early diagnosis and timely treatment of OSA.

## Figures and Tables

**Figure 1 brainsci-13-00838-f001:**
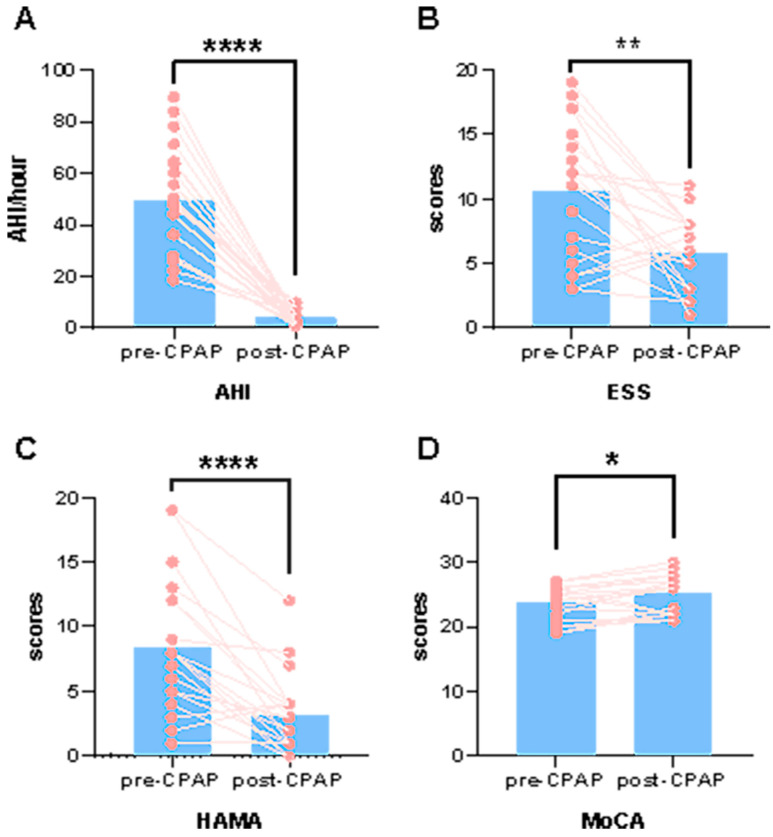
Clinical scores were changed between post-CPAP and pre-CPAP OSA patients. (**A**) The post-CPAP AHI (the post-CPAP AHI obtained from the ventilator data after the sixth month of treatment.), (**B**) ESS and (**C**) HAMA were significantly reduced compared to those in pre-CPAP OSA patients. (**D**) The post-CPAP MoCA scores were significantly increased compared to those in pre-CPAP OSA patients. * *p* < 0.05, ** *p* < 0.01, **** *p* < 0.0001.

**Figure 2 brainsci-13-00838-f002:**
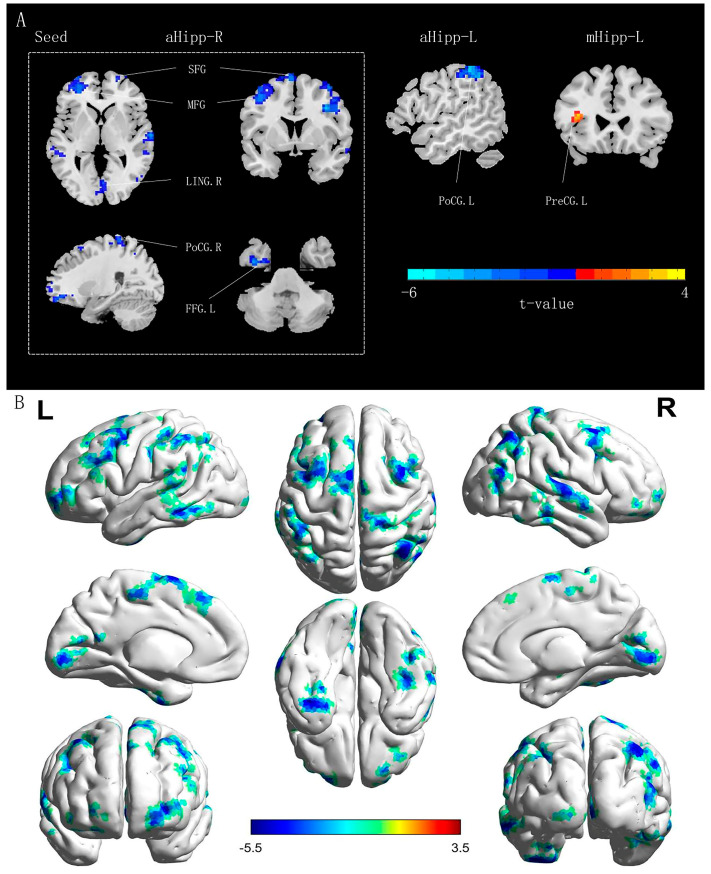
The abnormal FC in the hippocampal subregions of OSA: (**A**) all differential brain regions, and (**B**) including only the differential brain regions of the right anterior hippocampus as ROI. Warm colors (and cool) indicate significantly increased (and decreased) FC in the post-CPAP patients; All results were reported at voxel level *p* < 0.01, and cluster level *p* < 0.05, GRF corrected.

**Figure 3 brainsci-13-00838-f003:**
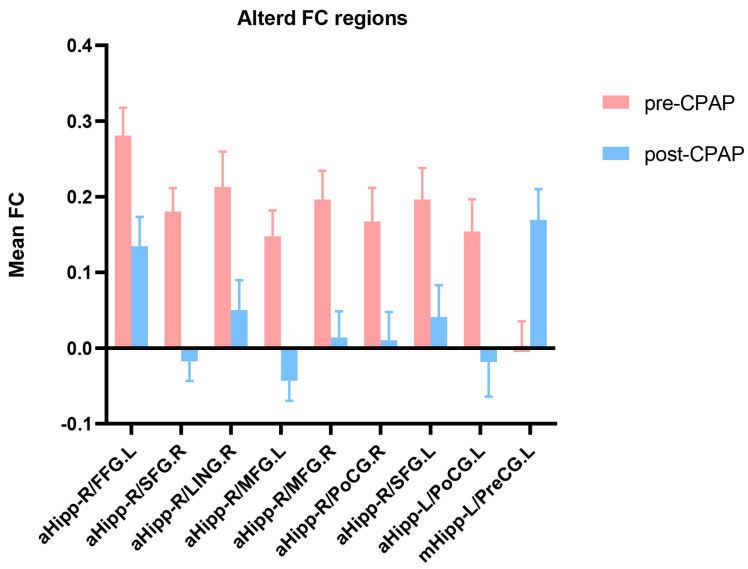
Mean weighted FC signal values of pre-CPAP and post-CPAP OSA patients in altered regional brain areas. FC, functional connectivity; pre-CPAP OSA, OSA patients before CPAP treatment; post-CPAP, OSA patients after CPAP treatment; FFG, fusiform gyrus; SFG, superior frontal gyrus; LING, lingual gyrus; MFG, middle frontal gyrus; PoCG, postcentral gyrus; PreCG, precentral gyrus; aHipp, anterior hippocampus; mHipp, middle hippocampus; L, left; R, right.

**Figure 4 brainsci-13-00838-f004:**
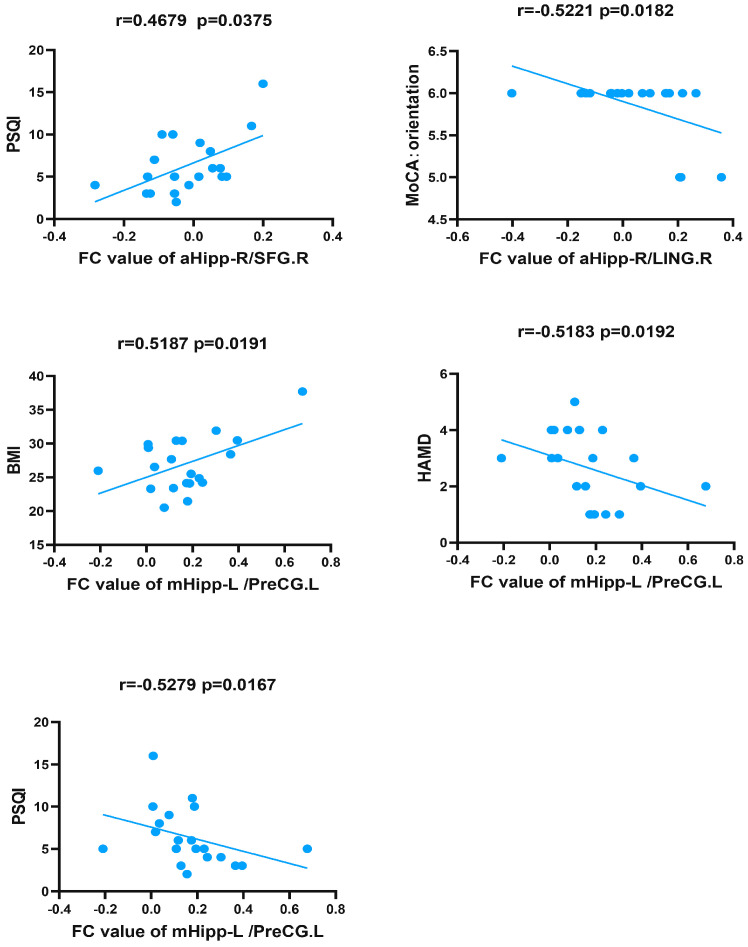
Correlations between hippocampal FC in post-CPAP patients and clinical variables in OSA. FC, functional connectivity; post-CPAP OSA, OSA patients after CPAP treatment; aHipp, anterior hippocampus; mHipp, middle hippocampus; L, left; R, right; SFG, superior frontal gyrus; LING, lingual gyrus; PrePG, precentral gyrus; BMI, body mass index; PSQI, Pittsburgh Sleep Quality Index; HAMD, Hamilton Depression Scale; MoCA, Montreal Cognitive Assessment.

**Table 1 brainsci-13-00838-t001:** Population and clinical characteristics of participants.

Characteristic	Pre-CPAP	Post-CPAP	*t*-Value/ *z*-Value	*p*-Value
	OSA Patients	OSA Patients		
	(N = 20)	(N = 20)		
Sex (male/female) ^a^	18/2	18/2	/	/
Age (years) ^a^	42.1 ± 8.2	42.1 ± 8.4	/	/
BMI (kg/m^2^) ^a^	27.3 ± 4.2	27.0 ± 4.1	0.808	0.429
Education (years)	12.2 ± 3.1	/	/	/
Nadir SaO_2_ (%)	94.3 ± 3.5	/	/	/
Mean SaO_2_ (%)	97.7 ± 1.8	/	/	/
SaO_2_ < 90%	0.2 ± 0.2	/	/	/
Sleep efficiency (%)	89.9 ± 2.8	/	/	/
AHI (events/h) ^a^	49.6 ± 21.3	4.0 ± 2.7	9.523	<0.001 ***
PSQI, scores ^b^	7.3 ± 4.0	6.4 ± 3.4	−0.700	0.484
ESS, scores ^a^	10.6 ± 5.2	5.8 ± 2.8	3.879	0.001 **
HAMA, scores ^a^	8.5 ± 5.1	3.2 ± 3.0	5.280	<0.001 ***
HAMD, scores ^b^	4.9 ± 2.7	2.7 ± 1.3	−2.562	0.010 *
MoCA, scores ^b^	23.8 ± 2.7	25.3 ± 3.1	−2.274	0.013 *
MoCA: visual space and execution ^b^	4.1 ± 1.1	4.7 ± 0.5	−2.221	0.026 *
MoCA: naming ^b^	2.9 ± 0.3	2.9 ± 0.3	/	1.000
MoCA: delayed memory ^b^	1.7 ± 1.2	2.5 ± 1.5	−2.984	0.003 **
MoCA: attentional function ^b^	5.2 ± 1.3	5.4 ± 1.0	−0.586	0.558
MoCA: language ^b^	2.3 ± 0.9	2.3 ± 0.7	−0.577	0.577
MoCA: abstract ^b^	1.6 ± 0.7	1.7 ± 0.6	−0.447	0.655
MoCA: orientation ^b^	5.9 ± 0.5	5.9 ± 0.4	/	1.000

Notes: ^a^ paired sample *t*-test and ^b^ Wilcoxon test; * *p* < 0.05, ** *p* < 0.01 and *** *p* < 0.001; pre-CPAP OSA, OSA patients before CPAP treatment; post-CPAP OSA, OSA patients after CPAP treatment; BMI, body mass index; SaO_2_, oxygen saturation; SaO_2_ < 90%, percentage of total sleep time spent at oxygen saturation < 90%; AHI, apnea hypopnea index; PSQI, Pittsburgh Sleep Quality Index; ESS, Epworth sleepiness scale; HAMA, Hamilton Anxiety Scale; HAMD, Hamilton Depression Scale; MoCA, Montreal Cognitive Assessment.

**Table 2 brainsci-13-00838-t002:** Brain areas showing significant differences in the FC of the hippocampal subregions between pre-CPAP and post-CPAP OSA patients (paired sample *t*-test).

Condition	Hippocampal Subregions	Brain Region	Voxel	MNI Coordinates of Peak Voxel			*t*-Value
				X	Y	Z	
Post-CPAP > Pre-CPAP	mHipp-L	PreCG.L	282	−36	24	15	5.325
Post-CPAP < Pre-CPAP	aHipp-R	FFG.L	171	−39	−9	−42	−5.897
		SFG.R	165	21	60	9	−5.335
		SFG.L	169	−3	6	69	−4.700
		MFG.L	158	−18	−81	33	−4.800
		MFG.R	307	3	−6	54	−5.261
		LING.R	217	3	−90	−3	−4.275
		PoCG.R	193	21	−36	72	−4.992
	aHipp-L	PoCG.L	181	−54	−36	54	−4.570

Notes: All clusters were reported with a voxel-level threshold of *p* < 0.01, GRF correction, and cluster-level of *p* < 0.05, two-tailed. FC, functional connectivity; pre-CPAP OSA, OSA patients before CPAP treatment; post-CPAP OSA, OSA patients after CPAP treatment; aHipp, anterior hippocampus; mHipp, middle hippocampus; L, left; R, right; PrePG, precentral gyrus; FFG, fusiform gyrus; SFG, superior frontal gyrus; MFG, middle frontal gyrus; LING, lingual gyrus; PoCG, postcentral gyrus.

**Table 3 brainsci-13-00838-t003:** Correlation analyses between altered FC of hippocampal subregions and clinical assessments in pre-CPAP and post-CPAP OSA patients.

Condition	FC Value of Brain Areas	Clinical Assessments	*r*-Value	*p*-Value
Post-CPAP OSA	aHipp-R/SFG.R	PSQI ^b^	0.468	0.038
	aHipp-R/LING.R	MoCA: orientation ^b^	−0.522	0.018
	mHipp-L/PreCG.L	BMI ^a^	0.519	0.019
		HAMD ^b^	−0.519	0.019
		PSQI ^b^	−0.528	0.017
	aHipp-R/FFG.L	AHI ^a^	−0.480	0.032
Pre-CPAP OSA	aHipp-R/SFG.R	Min SaO_2_ (%) ^a^	0.495	0.027
		Mean SaO_2_ (%) ^b^	0.561	0.010
	aHipp-R/LING.R	AHI ^a^	−0.472	0.036
		Min SaO_2_ (%) ^a^	0.553	0.011
		Mean SaO_2_ (%)	0.573	0.008
	aHipp-R/MFG.L	Min SaO_2_ (%) ^a^	0.593	0.006
		Mean SaO_2_ (%) ^b^	0.640	0.002
	aHipp-R/MFG.R	Min SaO_2_ (%) ^a^	0.447	0.048
		Mean SaO_2_ (%) ^b^	0.634	0.003
	aHipp-R/PoCG.R	AHI ^a^	−0.454	0.044
		Min SaO_2_ (%) ^a^	0.461	0.041
		Mean SaO_2_ (%) ^b^	0.496	0.026
	aHipp-R/SFG.L	HAMD ^b^	−0.445	0.049
		Min SaO_2_ (%) ^a^	0.450	0.047
		Mean SaO_2_ (%) ^b^	0.470	0.037

Notes: ^a^ Pearson correlation and ^b^ Spearman correlation analysis; FC, functional connectivity; pre-CPAP OSA, OSA patients before CPAP treatment; post-CPAP OSA, OSA patients after CPAP treatment; aHipp, anterior hippocampus; mHipp, middle hippocampus; L, left; R, right; SFG, superior frontal gyrus; LING, lingual gyrus; PrePG, precentral gyrus; FFG, fusiform gyrus; MFG, middle frontal gyrus; PoCG, postcentral gyrus; AHI, apnea–hypopnea index; PSQI, Pittsburgh Sleep Quality Index; HAMD, Hamilton Depression Scale; MoCA, Montreal Cognitive Assessment; SaO_2_, oxygen saturation.

## Data Availability

The original contributions presented in the study are included in the article. Further inquiries can be directed to the corresponding author.

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
