# Peer review of "Changes in Functional Connectivity of Hippocampal Subregions in Patients with Obstructive Sleep Apnea after Six Months of Continuous Positive Airway Pressure Treatment"

_brainsci, 2023, doi:10.3390/brainsci13050838_

Round 1
Reviewer 1 Report
This is a well-written paper that explores an important topic related to health and disease. Was this study powered? What is the sample size required? An RCT design would be required to adequately examine this research question and this should be acknowledged in the Discussion. There was not much mention of compliance with CPAP. This should be addressed considering the study uses a pre-post-study design. Some further comments are below.
Lines 115-117: A hypothesis is a proposed explanation for a phenomenon. This sentence does not make any proposal and therefore it is not a ‘hypothesis’. Need to state what you think will occur?
Line 162: Is the following correct “…5 others-evaluation”?
Line 199: Please be consistent and stick with ‘fMRI’ rather than ‘fMR images’. Check throughout the whole manuscript.
Line 325: Is the following an error “..orientation, orientation…”
The paper is generally well-written.
Author Response
Response to Reviewer 1 Comments
Point 1: Quality of English Language
Minor editing of English language required
Response 1: Thank you for your suggestion. We have asked relevant experts to carefully check the quality of the English language and make certain modifications. (https://www.editage.cn/, JOB CODE: LIMEF_1)
Point 2: Does the introduction provide sufficient background and include all relevant references?
Yes.
Response 2: Thanks.
Point 3: Are all the cited references relevant to the research?
Yes.
Response 3: Thanks.
Point 4: Is the research design appropriate?
Yes.
Response 4: Thanks.
Point 5: Are the methods adequately described?
Yes.
Response 5: Thanks.
Point 6: Are the results clearly presented?
Can be improved.
Response 6: Thanks. We have modified and improved the results to make them clearer.
Point 7: Are the conclusions supported by the results?
Can be improved.
Response 7: Thanks. We have revised the conclusions appropriately according to the results, and the conclusions are supported by the results.
Point 8: This is a well-written paper that explores an important topic related to health and disease. Was this study powered? What is the sample size required?
Response 8: Thank you for your affirmation. The main purpose of this study was to explore the changes of functional connectivity in the brain of the hippocampal subregions in OSA patients after CPAP treatment, and to provide a basis for the changes of neuroimaging in OSA patients after CPAP treatment. Due to poor compliance of OSA patients, relatively few subjects were included in this study, but many previous neuroimaging studies after CPAP therapy were also small. For example: (DOI: 10.1164/rccm.201005-0693OC; DOI: 10.1007/s41782-021-00190-0; DOI: 10.5665/sleep.3994 )
Point 9: An RCT design would be required to adequately examine this research question and this should be acknowledged in the Discussion.
Response 9: Thank you for your suggestion. We are fully aware of the need for a randomized controlled trial design to fully examine this research question, and it has been discussed that future studies will add control and alternative treatment groups.
Point 10: There was not much mention of compliance with CPAP. This should be addressed considering the study uses a pre-post-study design.
Response 10: Thank you for your suggestion. I have added this part to the revised manuscripts.
Point 11: Lines 115-117: A hypothesis is a proposed explanation for a phenomenon. This sentence does not make any proposal and therefore it is not a ‘hypothesis’. Need to state what you think will occur?
Response 11: Thank you for the reminder. It has been indicated in the revised manuscripts.
Point 12: Line 162: Is the following correct “…5 others-evaluation”?
Response 12: Thank you for the reminder. It has been indicated in the revised manuscripts.
Point 13: Line 199: Please be consistent and stick with ‘fMRI’ rather than ‘fMR images’. Check throughout the whole manuscript.
Response 13: Thank you for the reminder. It has been indicated in the revised manuscripts.
Point 14: Line 325: Is the following an error “..orientation, orientation…”
Response 14: Thank you for the reminder. It has been indicated in the revised manuscripts.
Comments on the Quality of English Language:
The paper is generally well-written.
Response: Thank you very much.

Reviewer 2 Report
The work is interesting and addresses an important topic, since Previous studies have shown that structural and functional impairments of hippocampal subregions in patients with obstructive sleep apnea (OSA) are related to cognitive impairment.
the proposed treatment may be favorable to reduce said damages.
Regarding improvements, in Table 2 I do not see the p value, that is, the significance, add, and the confidence interval.
Point out in limitations the lack of a control group or alternative treatment.
Author Response
Response to Reviewer 2 Comments
Point 1: Does the introduction provide sufficient background and include all relevant references?
Yes.
Response 1: Thanks.
Point 2: Are all the cited references relevant to the research?
Yes.
Response 2: Thanks.
Point 3: Is the research design appropriate?
Can be improved.
Response 3: Thank you for your advice. We have perfected and modified the design method in this paper. See the Methods section for details.
Point 4: Are the methods adequately described?
Can be improved.
Response 4: Thank you for your comments. We have revised and improved the method section.
Point 5: Are the results clearly presented?
Can be improved.
Response 5: Thanks. We have modified and improved the results section.
Point 6: Are the conclusions supported by the results?
Can be improved.
Response 6: We have revised the conclusions appropriately according to the results, and the conclusions are supported by the results.
Point 7: Regarding improvements, in Table 2 I do not see the p value, that is, the significance, add, and the confidence interval.
Response 7: Thank you for the reminder. It has been indicated in the revised manuscripts. (All clusters were reported with a voxel-level threshold of P < 0.01, GRF correction, and cluster-level of P < 0.05, two tailed. )
Point 8: Point out in limitations the lack of a control group or alternative treatment.
Response 8: Thank you for your suggestions. We have pointed this out in the limitations.

Round 2
Reviewer 1 Report
Well done on responding to the comments and improving the quality of your manuscript.
Has been improved.